# Improved and Innovative Accident-Tolerant Nuclear Fuel Materials Considered for Retrofitting Light Water Reactors—A Review

**Raul B. Rebak**

GE Research, Schenectady, NY 12309, USA; rebak@ge.com

**Abstract:** Since 2011, there has been an international effort to evaluate the behavior of newer fuel rod materials for the retrofitting of existing light water reactors (LWR). These materials include concepts for the cladding of the fuel and for the fuel itself. The materials can be broadly categorized into evolutionary or improved existing materials and revolutionary or innovative materials. The purpose of the newer materials or accident-tolerant fuels (ATF) is to make the LWRs more resistant to loss-of-coolant accidents and thus increase their operation safety. The benefits and detriments of the three main concepts for the cladding are discussed. These include (i) coatings for existing zirconium alloys; (ii) monolithic iron–chromium–aluminum alloys; and (iii) composites based on silicon carbide. The use of ATF materials may help extend the life of currently operating LWRs, while also being a link to material development for future commercial reactors.

**Keywords:** nuclear materials; LWR; FeCrAl; corrosion

## 1. Introduction

The world has been using electricity generated by harvesting the heat of reactions liberated during the fission of Uranium 235 since the 1950s, during which both the USSR and the USA connected light water reactors (LWR) to the civilian power grid. The first light water reactor to generate electricity for the civilian population was a pressurized type called Atomic Power Station 1 (APS-1) at Obninsk (USSR), which was connected to the grid on the 27th of June 1954 [1]. At approximately the same time, the Westinghouse company in the USA built a pressurized water reactor plant in Shippingport, PA, USA, which was connected to the civilian grid on 2 December 1957 [2]. Both pioneering plants did not generate much power, but they served an important function as test facilities for the advancement of materials in nuclear power technology. Since 1960, the USA has been using electricity generated by commercial light water reactors (LWR), including boiling water reactors (BWR) and pressurized water reactors (PWR). There are 94 LWRs producing power in the USA, and about two thirds of them are PWRs. The materials of construction in LWR remained practically the same for over sixty years. The reactor pressure vessel is made of carbon steel, and it may contain a 3 mm-thick layer of stainless steel weld overlay in the inside of the pressure vessel to provide protection against corrosion. Inside the pressure vessel, most of the components are made of austenitic stainless steel and chromium-containing nickel-based alloys [3]. The LWR fuel rod concept has not changed since it was designed in the late 1940s, always consisting of ceramic uranium dioxide fuel pellets protected by a metallic cladding or tube made of a zirconium (Zr)-based alloy [2,4]. The Zr alloy tube offers protection to the fuel from the coolant (water) and provides the surface for the removal of the fission heat from the rod by the water.

LWRs provide approximately 20% of the electricity in the USA. Nuclear energy is clean, and unlike other forms of carbon free energies such as wind and solar, nuclear is weather-independent and it can operate 24 h a day, seven days a week. The cost of energy from nuclear sources could be high compared, for example, to the burning of natural gas.

Currently, there is a plan at the federal level in the USA to prevent the premature shut down of operating nuclear power plants based on economics only. Nuclear energy could help the decarbonization efforts which are underway in many countries. The ATF materials that are discussed here are part of an effort to help retrofit the current LWRs for them to operate safely and more economically for a few more decades.

*The Excellent Performance of Zirconium Alloys as Fuel Cladding*

Since the 1940s, it has been known that if the temperature of the fuel rods in contact with water increases above 450 °C, the zirconium of the cladding would start oxidizing rapidly following an exothermic reaction [4–6]. Despite its narrow margin of temperature tolerance, Zr alloys (Table 1) have been used successfully by commercial nuclear power plants since the 1960s. Initially, the main alloying element for Zr alloys was less than 2% of Tin (Sn), but since the 1990s, Zr alloys containing a similar amount of Niobium (Nb) have been developed and implemented in the nuclear power stations worldwide (Table 1) [2]. The early Zr alloys had several environmental degradation issues with respect to both fuel cladding and BWR channels. The main degradation issues of Zr alloys in contact with the coolant were associated with general oxidation, nodular corrosion, galvanic corrosion, shadow corrosion, crud induced localized corrosion, hydriding, debris fretting, etc. [2,6,7]. The corrosion mechanisms of the Zr alloys were driven not only by the composition and microstructure of the used Zr alloys but also by poorly controlled water chemistry. Eventually, by cleaning the water of corrosion products (e.g., iron), adding hydrogen, zinc, and noble metal, most of corrosion degradation problems were understood and gradually retired. The only remaining major failure of Zr-clad fuel rods under LWR normal operation conditions is debris fretting [8,9]. Fretting is a process by which a foreign material (such as a stainless steel wire) repetitively touches the external surface of the cladding, causing its eventual perforation and allowing the coolant to enter into contact with the fuel and fission gases. Other failures issues, such as cracking from inside the cladding due to the accumulation of iodine, are now rare [2,7].

**Table 1.** Zr-based alloys used for fuel cladding.

| Alloy | Nominal Composition in Mass Percent |
|:---:|:---:|
| Zircaloy-2 R60802 | Zr + 1.2/1.7Sn + 0.07/0.20Fe + 0.05/0.15Cr + 0.03/0.08Ni (Fe + Cr + Ni = 0.18–0.38) |
| Zircaloy-4 R60804 | Zr + 1.2/1.7Sn + 0.18/0.24Fe + 0.07/0.13Cr (Fe + Cr = 0.28–0.37) |
| ZIRLO | Zr + 1Sn + 1Nb + 0.1Fe (Optimized Zirlo has 0.67Sn) |
| M5 | Zr + 1Nb + 0.14O |
| E110 | Zr + 1Nb |
| E635 | Zr + 1.2Sn + 1Nb + 0.35Fe |
| Zr-2.5Nb R60904 | Zr + 2.4/2.8Nb |

After the March 2011 Fukushima accident, where there was a large amount of hydrogen generated due to the fast oxidation of Zr components by water, the international nuclear materials community started to consider if more accident-tolerant materials could replace the well-known Zr alloys (Table 1). As such, the field of accident-tolerant fuels (ATF)—which later evolved into advanced technology fuels (ATF)—was founded to focus more upon the new technology and less upon the accidents. The main initial attributes of the ATF materials were that they should be more robust than Zr alloys, have lower reactivity with steam, and generate less hydrogen and heat than Zr. Figure 1 shows the main categories of accident-tolerant materials that have emerged for fuel rods [4,6,10,11]. The proposed materials solutions can be classified as (1) evolutionary or improvement of the current fuel architecture; or (2) innovative or revolutionary solutions, which involve materials never used before in LWRs. The ATF materials not only refer to cladding but also

to modifications for the fuel itself, such as doping the current uranium dioxide with chemicals to enhance its heat conductivity or to avoid a premature pellet fragmentation. Fuel producers are also investigating the use of other compounds of U for fuel such as silicide, carbide, and nitride. Other areas of associated studies involve increasing the concentration of U-235 in the pellets beyond 5% (higher enrichment) or extending the allowable burnup of the fuel beyond current regulatory restrictions of 62 GWd/MTU (Figure 1) [12]. Besides the fuel rods, other components that may need to be replaced in the reactor core using ATF candidates include BWR channel boxes and control rods.

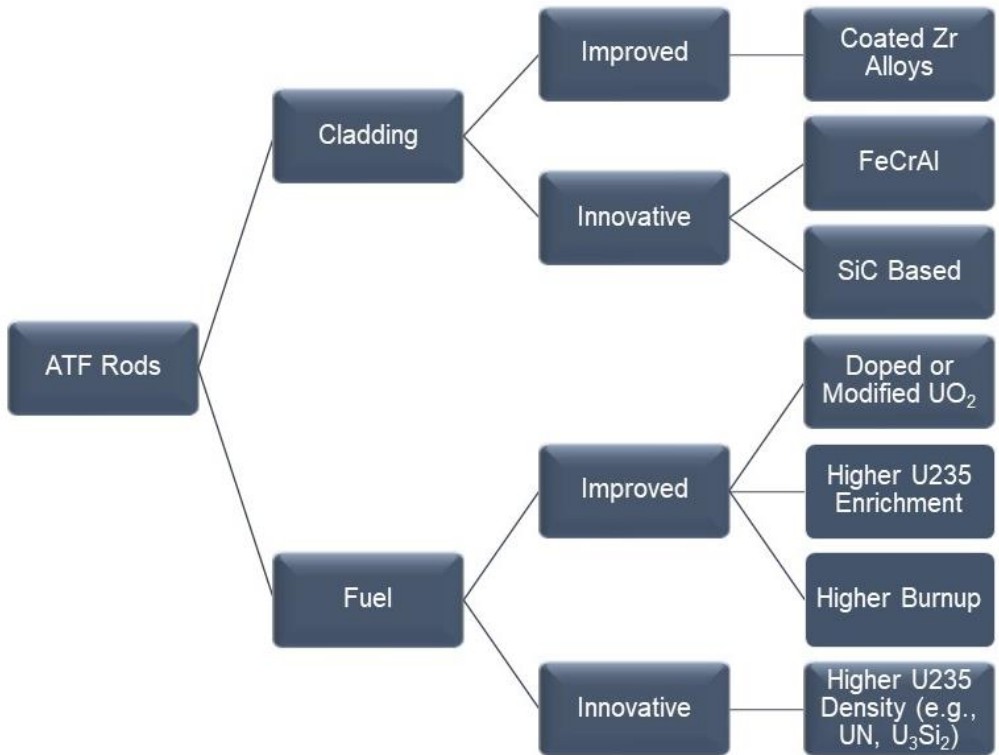

**Figure 1.** Accident-tolerant materials under consideration for LWRs.

The purpose of the current review is to outline the state-of-the-art knowledge of the properties of the cladding materials listed in Figure 1. This review will discuss not only the benefits or attributes of each type of material but also the concerns that their use may bring to the nuclear materials engineering community.

## 2. Improved and Innovative Materials for the Cladding

Figure 1 shows the three most common types of materials being investigated for the cladding of the fuel. The cladding materials are listed in the most likely chronological order in which they may be implemented (approved by regulatory authorities) in commercial power stations. This sequential order is directly related to our present knowledge on their likely performance in near 300 °C water under neutron irradiation. These newer LWR materials can be listed as follows: (1) coated Zr-based tubes (evolutionary or incremental gain, near term); (2) monolithic FeCrAl alloy tubes (revolutionary, metallic alloy, midterm); and (3) silicon carbide-based composites (revolutionary, a ceramic, longer-term implementation). These three materials are not new to the nuclear industry since all of them were investigated and researched for different nuclear applications before the concept of ATF was born in 2011. The technical attributes (benefits and detriments) of each of these three listed concepts are reviewed separately.

### 2.1. Coated Zr-Alloy, a Near-Term Concept (Evolutionary)

The coated Zr alloy ATF concept is considered evolutionary since the new product is a thin protective coating applied on top of Zr alloy rods already in use in LWRs [4,13]. This concept does not remove the Zr alloys from the reactor; the coating just makes these Zr alloys more robust with respect to environmental resistance during operating conditions and during upset or design basis accident conditions. The protective coating thickness would be in the order of 20 μm or less over a tube wall, which is in the order of 0.6 mm thick; that is, the coating will represent approximately 3% of the total wall thickness. The two primary objectives of the ATF coating are to protect against debris fretting under normal operation conditions (~300 °C) and to protect against attack by steam up to 1000 °C in the case of a loss of coolant accident (LOCA). The coating could also provide protection against sliding wear by contact interaction with tube separation grids during bundle fabrication; the coating could decrease the propensity towards ballooning by the Zr alloy substrate; and in the case of a burst, the coating could reduce the opening gap for fuel release and dispersion into the coolant. Another objective of the coating would be to minimize hydrogen entrance into the Zr alloy cladding wall, especially for extended burnups in the reactor core. This is possible since the coating would offer lower corrosion rates that the Zr alloy (thus generating less hydrogen) and because the coating itself could be a barrier to atomic hydrogen diffusion into the substrate.

All three LWR fuel vendors in the USA are currently working on developing coatings for the Zr alloy tubes. The coating characteristics may be different for PWR reactors than for BWR reactors since the environments are different. By its own architecture, a Zr alloy tubing with a coating has always been considered the most likely ATF concept to receive the earliest regulatory approval. For some fuel vendors, the coating developments for fuel rods started before March 2011—mainly to protect the rods against perforation via debris fretting both in PWRs and BWRs and against shadow corrosion at the grid locations in BWRs. The coating development projects were later incorporated under the ATF programs since it was found that the coating not only provided fretting resistance under normal operation conditions but also provided additional environmental protection compared to bare Zr alloy substrates.

Tang et al. [14] summarized the state of the art in coatings developments for ATF applications, discussing the wide range of coatings that were investigated globally by materials scientists. The coatings or surface transformation processes were grouped by families, namely, (1) surface modification by ion implantation; (2) non-metallic coatings such as carbon and silica; (3) metallic coatings such as chromium and FeCrAl alloys; (4) ceramic coatings such as alumina, MAX phases, carbides, nitrides, etc.; and (5) multi-layers coatings, each layer having only a few microns in thickness [14]. In general, for a coating to be viable for commercial fuel rods applications, it needs to be able to be applied economically on full-length cladding tubes (~5 m long) with the desirable architecture or composition, thickness, and microstructure [13]. The coating needs to be applied on Zr alloys at relatively low temperatures (e.g., below 400 °C) in order to preserve the properties of the Zr alloy substrate. The coating cannot add large neutron penalty; it needs to have a comparable coefficient of thermal expansion to the substrate in order to maintain adherence during thermal cycles and not have chemical interactions with Zr such as the formation of lower-temperature eutectic compounds. The merits of coating deposition methods—including (a) physical vapor deposition (PVD); (b) cold spraying; (c) cathodic arc evaporation; (d) magnetron sputtering; (e) electroplating, etc.—were addressed [14]. The technical review rated the coatings' survivability performance in contact with condensed water at approximately 300 °C (reactor normal operation conditions) and in superheated steam in the vicinity of 1000 °C (loss of coolant accident conditions). They reported, for example, that silicon and aluminum-based coating had an excellent performance in superheated steam, but these coatings performed poorly in condensed water (alumina and silica dissolve quickly in ~300 °C water). Other coatings based on nickel and titanium did rather well in subcritical water but were not satisfactory when exposed to steam. The

coating that rated the highest was chromium, which performed relatively well both in subcritical condensed water and in superheated steam [14].

In 2018, the Nuclear Energy Agency (NEA) of the OECD released a report on the state-of-the-art ATF including the concept of coatings for Zr alloys [13]. Two broad categories were identified by the NEA for the coatings being studied globally: (1) metallic coatings; (2) ceramic coatings. The following beneficial effects of coatings compared to bare Zr alloys were identified: (a) low neutronic penalty for coatings less than 20 μm thick; (b) similar mechanical behavior for the Zr alloy cladding if the coating is less than 20 μm thick; (c) higher resistance to environmental degradation both in water and steam for coated rods as compared to bare rods; (d) significant reduction to hydrogen pick up by the cladding; (e) increased wear resistance; (f) strengthening effect at higher temperatures with decreasing susceptibility to ballooning and creep [13]. The lower hydrogen pick-up during normal operation conditions of coated Zr alloy cladding may increase their performance in the back end of used fuel management, including transport, intermediate storage, and reprocessing [13].

In a review prepared for the US Nuclear Regulatory Commission (NRC), Geelhood and Luscher [15] discussed, more specifically, the degradation phenomena of Cr-coated Zr alloy cladding since this Cr coating seems the most likely candidate for PWR fuel applications. The leading deposition methods of Cr- on Zr-based alloys are physical vapor deposition (PVD) and cold spray. The authors gave an overview of the coating techniques used by the fuel vendors, identified the information available in the literature, and discussed the knowledge gaps for the Cr coatings, especially regarding to in-reactor irradiation performance [15]. Cr may react with Zr to form $ZrCr_2$ intermetallic compounds; however, at the normal operation temperature of the cladding (~350 °C), the formation of $ZrCr_2$ phases would not be significant to impair its performance.

During the initial coating development programs, most of the initial tests were performed in out-of-pile laboratory tests [16]. Results showed that M5 Zr alloy (Table 1) coated with Cr had no coating delamination and a delay on oxygen diffusion into the substrate under simulated accident conditions. Ramp tests showed smaller clad ballooning and smaller opening of the burst area, showing that there is a strengthening effect of the Zr alloy substrate via the thin Cr coating at higher temperatures [16].

It is crucial to demonstrate that these Cr coatings would endure in actual normal-operation-conditions commercial reactors environments. Fuel vendors have partnered with reactor owners' utilities to insert ATF-coated fueled elements into LWRs [17,18]. It has been reported that fuel rods of M5 Zr alloy coated with Cr through an optimized PVD process performed well during exposures to two 18-month cycles during in-pile irradiation at the PWR Vogtle Unit 2 power plant [17]. PWR coolants contain dissolved hydrogen gas, providing a reducing environment for the in-core metallic components such as the fuel rods. Pool-side inspections showed that the Cr coating provided significant reduction in oxidation kinetics in contact with the coolant in the reactor during normal operation conditions, leading to a reduction of hydrogen production and pick up by the cladding substrate [17].

Cold-spray-Cr-coated fuel rods of Zr alloy Zirlo (Table 1) were exposed to the Byron Unit 2 PWR environment for two cycles [18]. This insertion was first executed in 2019, when sixteen Cr-coated lead test rods (LTR) were introduced into the reactor core. Pool side inspection showed that after two cycles in the reactor, the hard-cold-spray Cr coating avoided scratching during fuel manufacturing and provided corrosion resistance, suppression of hydrogen pickup, and no indication of crud accumulation [18]. In parallel to the exposure in the Byron-2 nuclear power station, in June 2020, thirty-two cold-spray-Cr-coated lead test rods (LTR) were added to four 17 × 17 fuel bundles in peripheral positions at the Doel Unit 4 plant (Belgium) at the start of cycle 31 [19]. The enhanced visual inspection of the Cr-coated rods after Cycle 31 (in November 2021) showed clean surfaces, which were bright and uniform in their length. There was no delamination or cracking of the coating. While

the non-coated rods next to the Cr-coated segment showed some crud deposits in their lower section [19].

The use of Cr coating on Zr alloy cladding in PWRs may enable several economic benefits, such as (a) allowing for 24 months cycles through generating smaller burst openings which will improve performance regarding fuel fragmentation relocation and dispersal (FFRD); (b) allowing for a higher load of fissile uranium through a decrease in the cladding wall thickness to 0.4 mm because of the additional strength provided by the Cr coating; (c) allowing for plant up rates because of the extra strength and oxidation resistance provided by the coating during departure from nucleate boiling (DNB); and (d) allowing for more economical used-fuel management through stronger rods, as well as less oxidation or hydriding at the end of reactor residence, etc. [18].

The successful performance of a pure hard Cr coating in PWR primary circuit environments [17,18] does not necessarily translate to a satisfactory performance in BWR conditions, since the coolant in a BWR is more aggressive; that is, it contains oxidizing species generated by irradiation-induced hydrolysis that may accelerate the degradation of the coatings.

### 2.2. Monolithic FeCrAl, a Mid-Term Concept (Innovative or Revolutionary)

The second ATF cladding concept in Figure 1 is a monolithic tube of iron–chromium–aluminum (FeCrAl) alloy. This concept is innovative since it implies the radical exchange of one alloy for another non Zr-based alloy and because the ferritic FeCrAl alloy system has never been used in LWRs before. Since the outer diameter (OD) of the newer alloy would be the same as the current Zr alloy, the coolable surfaces and thermal hydraulic conditions will practically remain the same. The idea of a steel-like alloy for cladding is not completely new since in the early 1960s, some power reactors used cladding of 300 series austenitic stainless steels despite their higher neutron absorption cross section. As the price of Zr alloys became more affordable for the industry, and due to in-service cases of stress corrosion cracking at the welds of the stainless steel clad rods, the idea of an austenitic stainless steel for the cladding of the fuel was eventually abandoned [2].

The main reason FeCrAl alloys are currently being considered for ATF cladding is because of their high-temperature resistance to oxidation in air and steam as compared to Zr alloys [4,13,20–22]. Table 2 shows a short list of the most significant FeCrAl alloys under study for ATF cladding applications. Before the concept of ATF was born, FeCrAl alloys have been considered in the 1950s and 1960s for several nuclear applications [23,24]. Table 3 shows the main advantages of using FeCrAl for ATF cladding as compared to the well-known behavior or properties of the Zr-based alloys. In the next sections, some of these characteristics will be addressed in more detail.

**Table 2.** FeCrAl alloys for ATF fuel cladding.

| Alloy | Nominal Composition in Mass Percent |
| :---: | :---: |
| APM | Fe-21Cr-5.8Al |
| APMT | Fe-21Cr-5Al-3Mo |
| C26M | Fe-12Cr-6Al-2Mo |
| NFD ODS | Fe-12Cr-6Al |
| Aluchrom YHf | Fe-19Cr-5Al |

**Table 3.** Characteristics of FeCrAl alloys' cladding as compared to Zr alloys.

| Attributes and/or Benefits | Detriments and/or Concerns |
|---|---|
| 1. Outstanding resistance to oxidation by air and steam to temperatures near 1500 °C (melting);<br>2. Low coefficient of thermal expansion and high thermal conductivity;<br>3. Resistant to stress corrosion cracking from the coolant side; resistant to cracking from fuel cavity side by fission gases;<br>4. Excellent mechanical properties well above cladding operation temperatures;<br>5. Resistant to ballooning and small burst opening at failure;<br>6. As bcc materials, they are resistant to radiation damage such as swelling;<br>7. They are nickel-free, decreasing the danger of activation and decreasing their cost;<br>8. Inexpensive cladding fabrication methods by powder metallurgy extrusion and tube pilgering;<br>9. Low uniform corrosion rate from the coolant side at normal operation conditions; zinc water chemistry lowers the corrosion rate even more;<br>10. Resistant to galvanic corrosion and shadow corrosion in contact with grid separators;<br>11. Resistant to debris fretting. | 1. Lower transparency to thermal neutrons than Zr;<br>2. Likely increase in tritium content in the coolant;<br>3. Possible embrittlement due to neutron irradiation. |

### 2.2.1. Resistance to Oxidation in Steam at T > 1000 °C

FeCrAl alloys have been in use in the industry since the 1930s because of their strength and resistance to oxidation in air and steam. One alloy that has received attention recently regarding ATF application is APMT (Table 2). These Fe-based alloys contain enough Cr and Al to make them resist oxidation up to their near melting point at approximately 1400–1500 °C. In these alloys, Cr and Al act synergistically; Cr provides protection to approximately 1000 °C, and above this temperature, the protection is given by aluminum oxide [4,20,25–27]. Pint et al. [20] exposed coupons of several alloys to a 3.4 bar of 100% steam for 8 h from 800 °C to 1350 °C and determined the mass change of the coupons. At 1000 °C, the Zircaloy-2 coupon had a mass gain of 48 mg/cm$^2$, while the mass gain for APMT was less than 0.5 mg/cm$^2$. The high Cr (>25%Cr) ferritic alloys without aluminum (such as Fe29Cr4Mo) showed substantial degradation above 1200 °C, showing that Cr alone cannot protect the alloy in steam [20]. For example, at 1200 °C, the mass change for 29%Cr AL29-4C was 4 mg/cm$^2$, while for APMT, the mass change was less than 0.5 mg/cm$^2$ due to the formation of a solid-state alumina layer which is a diffusion barrier for oxygen from the environment [20]. This surface layer of alpha-alumina is practically inert to the presence of steam and does not allow oxygen to ingress and initiate internal oxidation of APMT [20]. Pint et al. [28] published results on the oxidation resistance in steam of candidate materials for cladding. Their goal was to identify materials that would have a factor of 100× lower in the oxidation rate constant than the current Zr alloys at 1200 °C. For the steel alloy family, they reported that the high Cr but aluminum free austenitic type 310SS (Fe20Ni25Cr) did not meet this oxidation rate reduction goal, while APMT (Table 2) exceeded the goal by a factor of 10×. The composition of the FeCrAl is also important for steam attack resistance. For example, 4 h exposure tests to 100% steam showed that APMT (Table 2) had a protective alumina until a temperature of 1475 °C, while a non-commercial alloy with 15%Cr and 5Al had rapid oxidation above 1350 °C [28]. Lipkina et al. [29] and Sakamoto et al. [30] tested the performance of the NFD ODS alloy (Table 2) in air and steam at temperatures in the vicinity of 1200 °C, and they found similar low oxidation rates, as published by Pint et al. [28]. It was reported that the parabolic oxidation rates of NFD ODS were practically the same in air and in steam, even though the thickness of the alumina layer was slightly thicker in steam than in air. This difference in thickness between steam and air was attributed to the earlier formation of the protective alpha alumina layer in the steam environment [29].

Rebak et al. [31] exposed coupons to APMT and C26M (Table 2) for 2 h and 4 h to 100% steam and to laboratory air from 800 to 1300 °C. They reported that both alloys had similar oxidation resistance in both environments, but the dependence of oxidation with temperature was different for air and steam. In the lower temperature range, the mass gain due to oxidation was faster in steam; but in the higher temperature range, the oxidation was faster in air [31]. Figure 2 shows that the resistance to oxidation of APMT was provided by a thin—and highly adherent to the substrate alpha—alumina film. Figure 2a shows that the alumina film in air was approximately 2.5 μm thick and it was columnar near the substrate and had equiaxed grains in the outer surface. Figure 2b shows that in steam, the film was approximately 2 μm thick and composed only of columnar grains; it is apparent that the external smaller grains (which are present in air) were removed by interaction with steam.

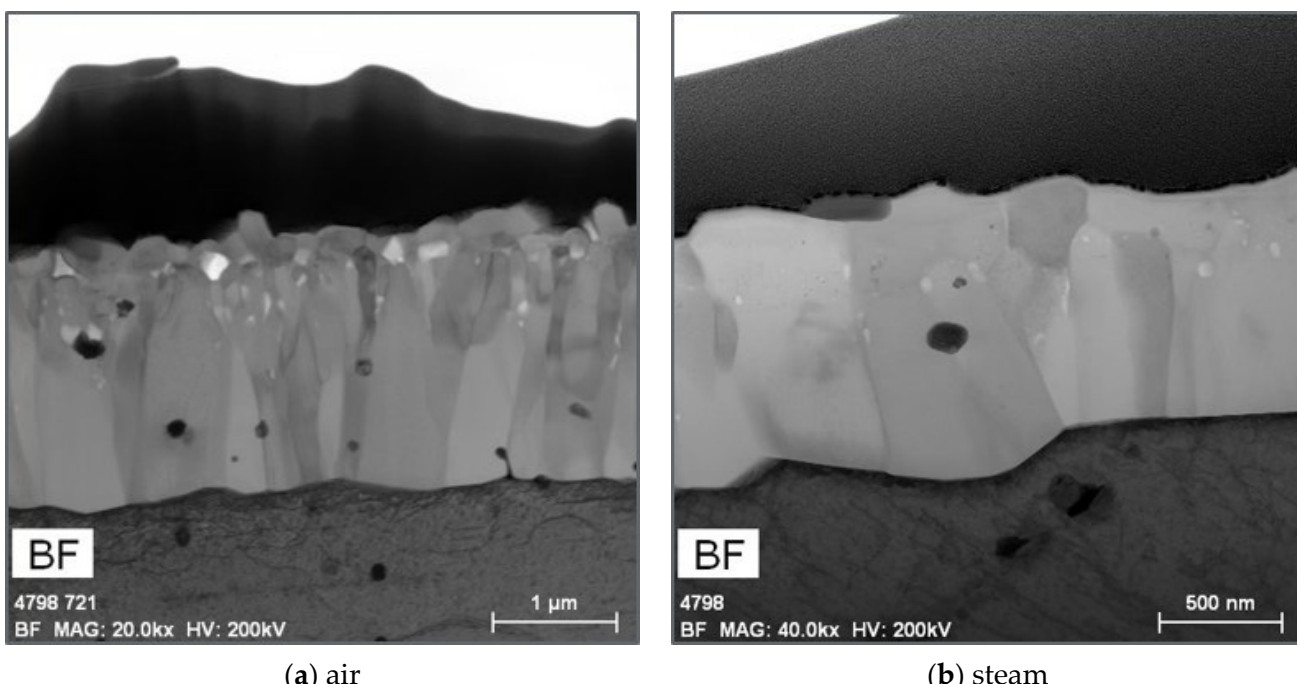

(**a**) air  (**b**) steam

**Figure 2.** Alpha alumina film on the surface of APMT coupons exposed at 1200 °C for 4 h in (**a**) air and (**b**) steam [31].

### 2.2.2. FeCrAl Alloys Are Resistant to Stress Corrosion Cracking from the Water and Fuel Side

It is difficult to address in detail all the attributes of the FeCrAl alloys for cladding applications (Table 3). The outstanding mechanical properties of FeCrAl, such as APMT compared to Zircaloy-2, have been recently discussed elsewhere [32]. The fact that APMT is much stronger than Zirc-2 at temperatures near 350 °C would allow for the reduction of the wall thickness of the APMT to 0.3 mm from the 0.6 mm currently used for Zr alloys. This reduction in wall thickness will compensate for the higher thermal neutron absorption of the FeCrAl alloys compared to Zircaloy. APMT and other FeCrAl alloys have excellent mechanical properties and creep resistance at temperatures in the order of 800 °C, which is more than double the temperature of the normal operation of the cladding. APMT and the newer generation of C26M cladding tubing are fabricated using powder metallurgy, which results in small grain sizes and high yield strength [32]. APMT has small additions of other alloying elements, developing nanosized particles that further increase its strength [28]. FeCrAl alloys also have a higher resistance to creep than Zircaloy-2, which is going to provide a delay to ballooning or rod burst, and in general, the lower creep rate will help maintain better dimensional stability in case of a design basis accident [4,27].

Austenitic stainless steels are susceptible to stress corrosion cracking (SCC) in LWR environments [33,34]. Ferritic stainless steels are less prone to SCC in LWR environments than their austenitic cousins [35–37]. The resistance to cracking of ferritic stainless steels was also reported in supercritical pressurized water pertaining to Generation IV reactors [38,39]. Not only ferritic Fe–Cr alloys are resistant to cracking; ferritic FeCrAl alloys such as APMT and C26M (Table 2) are also resistant to SCC in LWR environments, both under hydrogen and oxygen conditions [36,40]. Figure 3 shows the crack growth rate in compact specimens as a function of the applied stress intensity in oxidizing BWR-type environments. Austenitic type 304SS steels have a crack growth rate of $3 \times 10^{-7}$ mm/s at a stress intensity of 25 ksi$\sqrt{\text{in}}$, while ferritic stainless steel may only undergo minimal cracking at stress intensities higher than 40 ksi$\sqrt{\text{in}}$ [40].

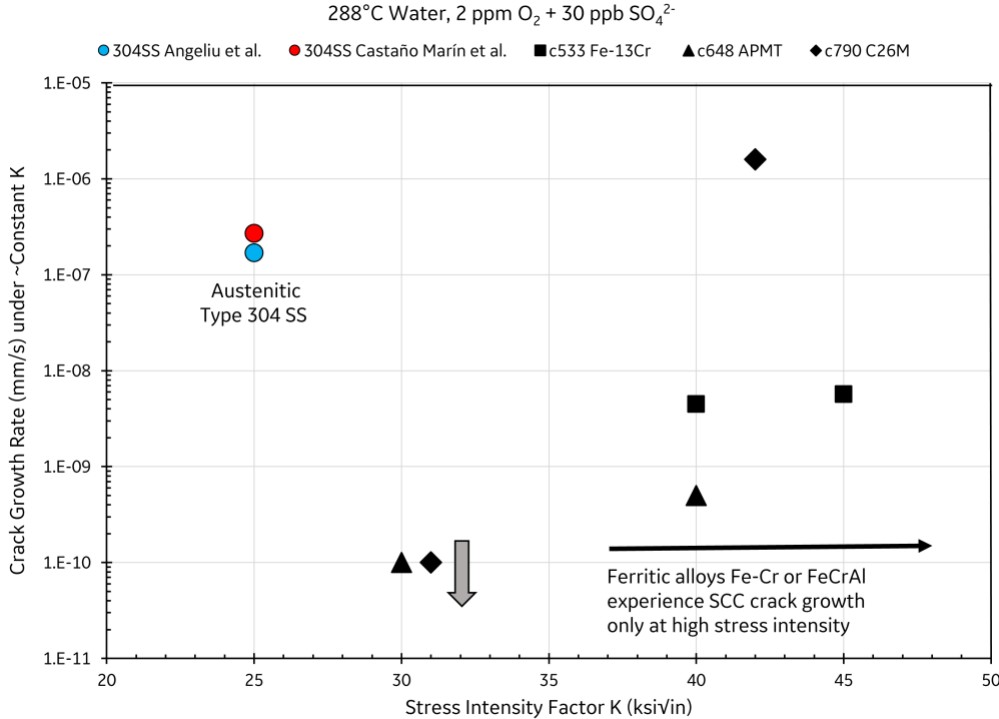

**Figure 3.** Susceptibility of austenitic and ferritic stainless steels to SCC in LWR environments [40].

It has been known since the 1960s that Zr-based alloys may suffer from stress corrosion cracking from the fuel cavity due to the presence of the element Iodine (I), which is the result of fission reactions [41]. The SCC promoted by iodine from the ID of the Zr alloy cladding tube is generally recognized as the result pellet–cladding interaction (PCI), which would impart tensile stresses to the internal diameter (ID) of the tube. The issue of cracking of the Zr alloy due to exposure to iodine is still being studied today even though plant rod failures related to this mechanism are rare. Since FeCrAl alloys are being investigated to replace Zr alloys as the cladding, it was important to examine if the new alloys would undergo cracking in the presence of fission products as well. Sakamoto et al. [42] performed testing of NFD ODS (Table 2) tubing (plugged at both ends) containing iodine at 350 °C. A Zircaloy-2 tube (no liner) was tested using the same conditions. They reported cracking in the Zircaloy-2 tube at an ID surface plastic strain between 0.012 and 0.018 and no cracking for the FeCrAl NFD ODS tube with a higher plastic strain between 0.015 and 0.021 [42]. This was the first time that it was reported that FeCrAl alloy cladding would be resistant to cracking from the fuel cavity side in the presence of fission products. Sakamoto et al. [43] also explored the effect of Cesium (Cs) on the cracking susceptibility of FeCrAl NFD ODS in a similar set up as was used for the Iodine test [42]. They reported cracking only when the partial pressure of Cs was 24 Pa and 800 Pa [43]. There was no cracking of FeCrAl after

72 h of testing when the partial pressure of Cs was 0.2 Pa. Under normal operation conditions, the partial pressure of Cs inside a fuel rod cavity would be in the order of $10^{-8}$ Pa [43].

2.2.3. FeCrAl Alloys Undergo Uniform Passive Corrosion in Normal Operation Conditions

When designing corrosion control, it would be better to select a material that does not corrode. Since this is not likely, the second-best option is to select a material that would recede uniformly, incorporating corrosion allowance into the design. Localized corrosion such as pitting corrosion or intergranular attack are not desirable since they could be sites for stress concentration. The corrosion characteristics of Zr alloys in condensed water at near 300 °C has been studied for many decades. It is known that when Zr oxidizes in water, it develops an adherent $ZrO_2$ oxide on the surface via the inward diffusion of oxygen anions. This oxide may start spalling after a certain threshold oxide thickness (e.g., 50 to 100 µm) is achieved [7,44].

Since 2013, many studies of corrosion of FeCrAl alloys have been conducted in out-of-pile simulated LWR conditions, such as pure water in the vicinity of 300 °C containing either oxygen or hydrogen dissolved gases. While Zr alloys show weight gain in both oxygen- and hydrogen-rich water, FeCrAl generally loses some mass in hydrogenated or reducing conditions and may gain mass in oxygenated or oxidizing conditions [45–48]. This behavior is understandable since the corrosion potential of APMT, for example, increases more than 700 mV going from a hydrogen water chemistry to an oxygen water chemistry [49]. The amount of mass loss or gain for FeCrAl will depend mostly on the amount of Cr in the alloy [48]. In hydrogen atmospheres, the oxide film on the surface of the FeCrAl coupons consisted mainly of a thin layer of chromium oxide ($Cr_2O_3$), while in oxygen containing waters, the oxide consisted or a double layer, i.e., an external thicker layer with a mixture of iron and chromium oxides and an internal layer of nearly pure $Cr_2O_3$. After extended immersion times in the autoclave (e.g., one year), an enrichment of aluminum develops underneath the $Cr_2O_3$ layer [48]. This is a similar mechanism as in steam oxidation [50], albeit with much slower kinetics since in steam at 1200 °C, the alumina layer may form in a few seconds, while in condensed water at 300 °C, the aluminum layer may take months to develop.

Figures 4 and 5 show the mass change for open tube specimens of Zirc-2, C26M, and APMT as a function of immersion time in out-of-pile autoclave systems. Figure 4 is for hydrogen water chemistry (HWC) and Figure 5 is for normal water chemistry (NWC) or containing oxygen. Under hydrogen chemistry, both FeCrAl coupons (C26M and APMT) lost a considerable amount of mass, and the higher mass loss of up to 100 mg/dm$^2$ corresponded to C26M or the alloy with 12%Cr. In HWC, the mass loss for APMT (21%Cr) was approximately 20 mg/dm$^2$. For the NWC autoclave, the mass loss for the two FeCrAl tube coupons was minimal or less than 10 mg/dm$^2$. Under HWC, the oxide film is only a 10 nm thick layer of $Cr_2O_3$. This oxide does not have an external oxide layer based on Fe to protect it from the environment. Therefore, it is speculated that the $Cr_2O_3$ layer may slowly recede in time while the layer continues to grow in the region in contact with the metal. In the case of NWC (Figure 5), there is no obvious recession because the external layer of Fe-rich oxide protects the underlying layer of $Cr_2O_3$; hence, the mass loss is insignificant.

Figures 4 and 5 show that of the two FeCrAl cladding materials, the recession of C26M was higher than the recession for the APMT material. The reason for this is the higher concentration of Cr in the APMT material, which helps to maintain a better passivation (Table 2). Most light water reactors currently use Zinc chemistry; that is, the coolant in the reactor has dissolved zinc in the parts per billion range with the objective of reducing corrosion and minimizing crud deposition on the fuel rods. It was shown that a few parts per billion Zn reduced the corrosion rate of FeCrAl, C26M, and APMT by 60% and reduced the oxide film thickness on the surface by three to four times [51].

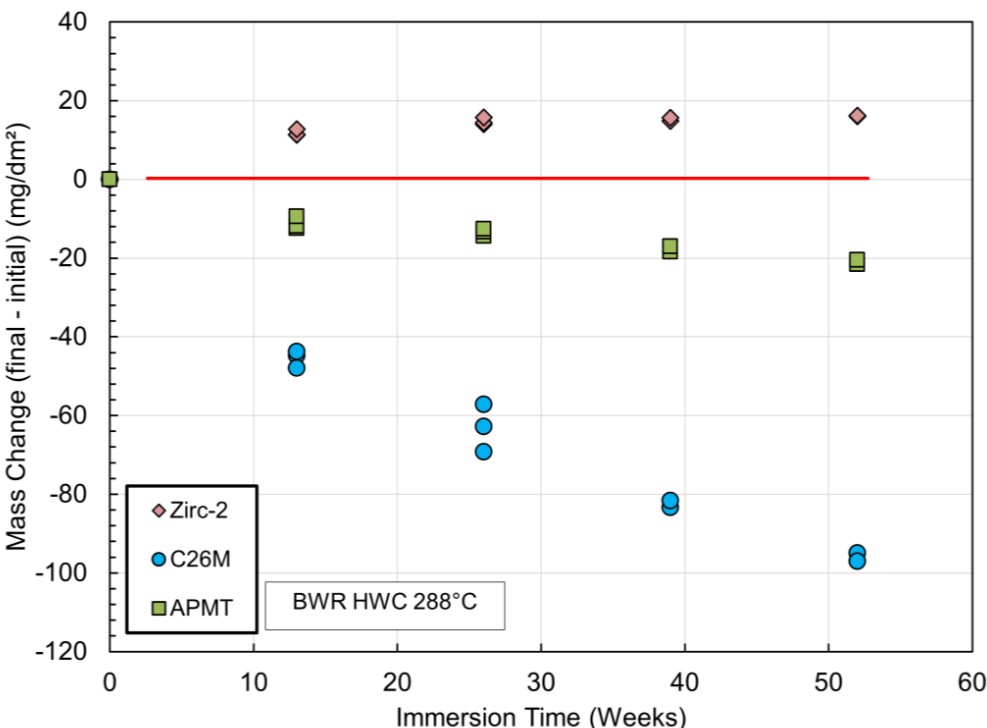

**Figure 4.** Mass change as a function of immersion time for Zirc-2, C26M, and APMT tube coupons in pure water at 288 °C containing 300 ppb of hydrogen gas (HWC) [48].

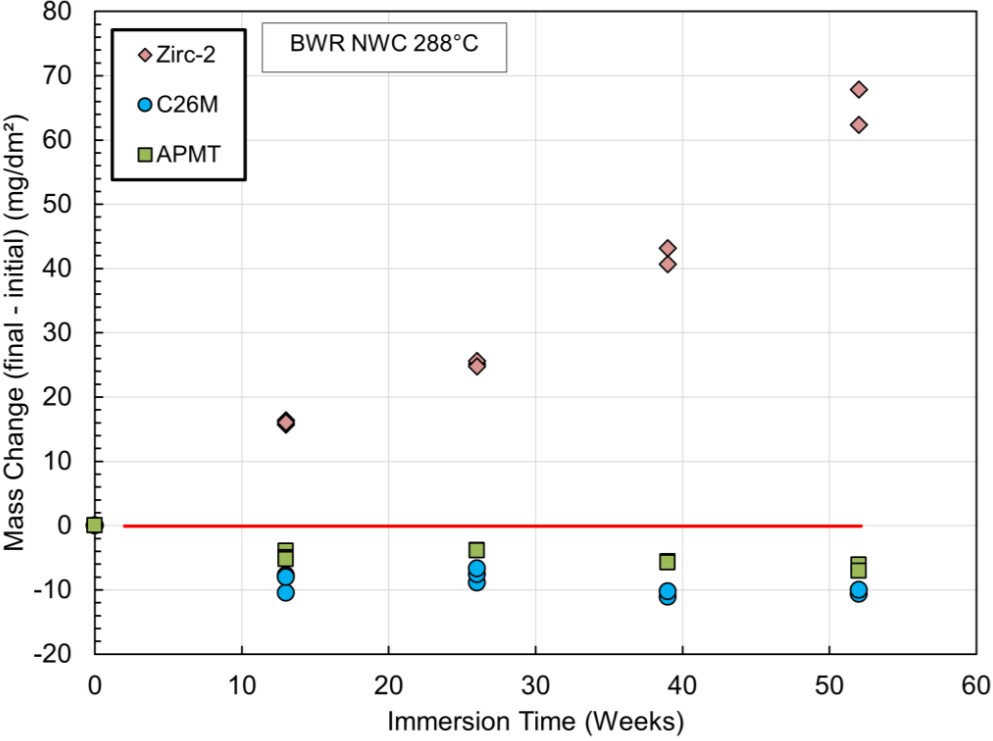

**Figure 5.** Mass change as a function of immersion time for Zirc-2, C26M, and APMT tube coupons in pure water at 288 °C containing 2 ppm of oxygen gas NWC) [48].

Because in high temperature water FeCrAl develops a surface oxide rich in Cr, it does not generate an electrochemical potential difference with the grid spacers or other stainless components in the reactor core that could lead to galvanic corrosion or shadow corrosion [49].

2.2.4. FeCrAl Alloys Are Resistant to Debris Fretting

Currently, the major fuel rod failure in operating LWRs is the perforation of the protective cladding due to fretting wear [8,9]. Once the cladding is breached, the coolant may enter into contact with the fuel and the fission gases, allowing for radioactive elements to escape into the coolant. One of the main reasons (even before the events at Fukushima) that the nuclear materials industry was exploring the possibility to using coatings to protect Zr-based alloys cladding was because the hard coating would protect against fretting wear of the cladding. This fretting wear could be a result of grid-to-rod or by foreign debris. If a monolithic FeCrAl alloy cladding is used in place of the current Zr-based alloy cladding, it may seem intuitive, based on their relative mechanical properties in the vicinity of 300 °C, that the FeCrAl would offer better wear resistance than the Zr alloys [26,52].

The relative wear resistance of APMT and Zircaloy-2 against the nickel-based alloy X-750 was assessed in moist air at 300 °C [53]. In this test, discs or pucks of the alloys were used. While the X-750 puck was static, the pucks of either APMT or Zircaloy-2 were rotating against the X-750 puck at 50 rpm for a total testing time of 20 h. The wear damage was evaluated by characterizing surface wear scars. Results showed that there was considerable Zircaloy-2 material transported to the X-750 puck. For the APMT puck case, some material from X-750 was transported to the APMT puck. Results show that in the case of a X-750 grit separator, its vibration against a Zircaloy-2 rod will cause the rod to recede or wear, but in the case of a APMT rod, the recession or wear will happen at the grid.

Sakamoto et al. [42] tested the fretting resistance of FeCrAl NFD ODS (Table 2) using (a) the conventional constant contact mode and the (b) impact fretting or normal oscillatory mode. Two types of stainless steels balls (304SS and 440C) were used as the contact material. The initial hardness of the 304SS was approximately 260 HV, and the hardness of the ferritic 440C steel was approximately 800 HV. The microhardness of the FeCrAl NFD ODS was approximately 290–320 HV, and that of Zircaloy-2 was 160–190 HV. The contact length was 35 μm, applied with a frequency of 10 Hz and a force of 2 N. After $2 \times 10^5$ cycles it was reported that if the 440C ball was used in impact mode, the wear volume loss of Zirc-2 was higher than $2.5 \times 10^{-4}$ mm$^3$ compared to a volume of less than $1 \times 10^{-4}$ mm$^3$ for FeCrAl ODS [42]. Other tests at ambient temperature contacting both FeCrAl ODS and Zircaloy-2 against 304SS—both in air and in water—using 19.6 N contact force on 1 mm in length showed higher resistance to sliding wear of FeCrAl ODS under both dry and wet conditions [54].

Figure 6 shows the results from a wear test conducted for two weeks in an autoclave with BWR NWC conditions or pure water at 288 °C containing 1 ppm dissolved oxygen [6]. In this test, a protruding 0.031-inch (0.79 mm) diameter spring wire of type 304SS was fixed on a rotating central shaft, and as the central shaft rotated, the wire touched repeatedly four vertical cladding tubes. These tubes were at 90° from each other. After 14 days of testing, the wear mark or groove on the Zirc-2 tube was approximately 600 μm deep (through the wall) and with a width of more than 1 mm (Figure 6a,b). In this side-by-side test, the groove mark on the APMT-2 tube was less than 100 μm deep and less than 1 mm wide (Figure 6c,d). This test, performed in simulated BWR environments, shows the superior resistance of APMT cladding to Zircaloy-2 cladding.

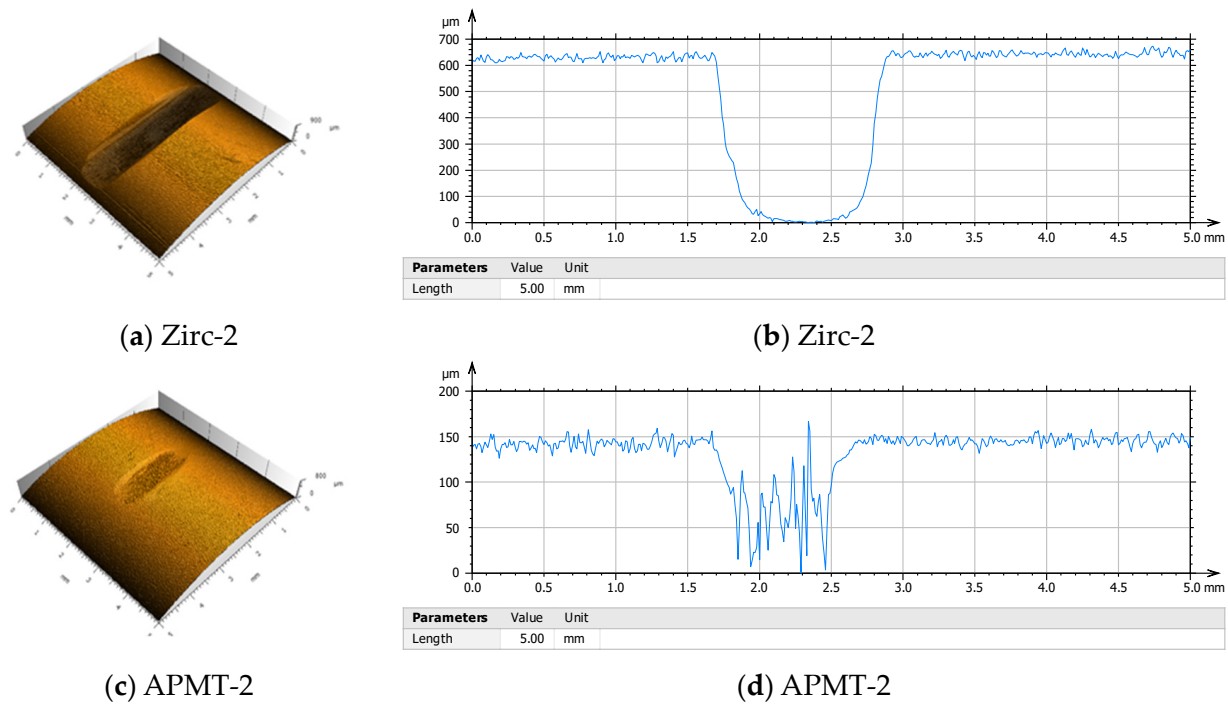

**Figure 6.** Relative wear rate of APMT and Zircaloy-2 after a test wherein a rotating wire of 304SS would repetitively touch the tubes. (**a**,**b**) A 600 µm deep wear groove formed on the Zirc-2 cladding after two weeks testing in 288 °C water. (**c**,**d**) A shallower 100 µm deep groove formed on the APMT tube after two weeks in 288 °C water [6].

Regarding the less desirable attributes (Table 3), FeCrAl alloys are expected to be less transparent to thermal neutrons than Zr-based alloys [6]. By reducing the wall thickness of the cladding from 0.6 mm for Zr alloys to 0.3 mm for FeCrAl (APMT), there is going to be a neutral (no cost) effect regarding parasitic neutron absorption. Another less desirable attribute is the issue of tritium release into the coolant. For a cladding application, Zr alloys do not release tritium into the coolant because tritium and hydrogen become captured by the cladding itself, developing hydrides. FeCrAl alloys are more transparent to hydrogen (tritium) than Zr, and none of its elements (Fe, Cr or Al) react with H to form hydrides; therefore, it was anticipated that more tritium may be released into the coolant when a FeCrAl cladding is used [22,55]. It is argued that the formation of surface oxides on the walls of the FeCrAl cladding could be a barrier for tritium release into the coolant [22,26,56].

### 2.2.5. Irradiation Campaigns and Deployment

Another concern regarding the use of ferritic alloys containing Cr is the embrittlement effect due to alpha prime formation. Thermal aging studies have shown that there is a reduction in ductility due to alpha prime formation in APMT; however, there is a remaining acceptable ductility in the material of approximately 10% [57]. The effects of neutron irradiation on alpha prime formation and overall radiation damage such as swelling need to be studied in more detail [26].

APMT rodlets were neutron irradiated at the Idaho National Laboratory (INL) Advanced Test Reactor (ATR) in two campaigns, namely, ATF-1 and ATF-2. In the first campaign, the rodlet ATF-8 (or G03) containing GNF UO$_2$ fuel was dry irradiated to 18.3 GWd/MTU. Post-irradiation examinations (PIE) showed no significant changes in the hardness of APMT cladding due to the neutron irradiation [22,58]. Moreover, there was no degradation of the APMT cladding from the fuel cavity due to interaction with the fuel [59]. In a second campaign, ATF-2, rodlets of both C26M (traditional metallurgy) and APMT (powder metallurgy) containing GNF UO$_2$ fuel were introduced into the ATR with their OD surfaces in contact with the coolant. These rodlets were removed in June 2021 after

receiving a dose of approximately 20 GWd/MtU. Pool-side optical examination of a C26M ATR ATF-2 rodlet showed no apparent corrosion from the OD.

Fueled and non-fueled articles of both APMT and C26M were also inserted into two commercial reactors: (a) the Hatch Power Plant #1 in February 2018; (b) the Clinton Power Plant #1 in October 2019. The IronClad articles survived the residence in the reactor and were transported to cooling pools. Currently, Oak Ridge National Laboratory (ORNL) is working on post-irradiation examination (PIE) studies of iron-clad articles removed from Hatch NPP. The Clinton rodlets are listed to be sent to ORNL in early 2024 for PIE studies.

In cases where the current ferritic versions of APMT are not found suitable after in-situ irradiation exposure evaluations, the alloy may be modified slightly to obtain a duplex or even a fully austenitic microstructure that could be more resistant to neutron irradiation embrittlement and alpha prime formation.

Since the FeCrAl ferritic alloys have excellent mechanical properties (Table 3) and oxidation resistance at temperatures in the order of 800 °C and higher [32], they are also being considered for application in future Generation IV reactors and fusion reactors.

### 2.3. Silicon Carbide-Based Cladding—A Long Term Concept (Revolutionary)

The third ATF cladding concept in Figure 1 is a cladding tube based on a composite of the ceramic compound silicon carbide (SiC). The SiC-based concept is truly revolutionary since it is not a coating on an existing product and it is not a monolithic alloy like a stainless steel. The basic architecture of the ceramic cladding would be a tube made of SiC fibers and then impregnated with a SiC matrix. This is necessary since the SiC itself is brittle and it needs the strong fibers to reinforce it. The fracture toughness of the composite is higher than the one for the matrix by itself [60]. As for FeCrAl, the idea of using SiC for nuclear applications is not new since it was explored before the ATF concepts were born in 2011 [4,60,61]. The use of a SiC composite may allow the cladding to stand temperatures up to near 1700 °C during accident conditions [17].

Figure 7 shows the attributes or benefits and detriments or concerns regarding the use of a SiC composite for an ATF cladding. The main attributes of SiC for cladding applications are their transparency to neutrons, their resistance to neutron irradiation damage, and their outstanding resistance to attack by steam. The slow reaction with steam will not produce ignitable hydrogen gas, which is one of the main concerns that led to the ATF program [4,17,28,60,62]. Coupons of chemical-vapor-deposited (CVD) SiC were tested from 2 h to 48 h in 100% steam and in a mixture of hydrogen gas with 50% of $H_2O$ at temperatures ranging from 800 °C to 1350 °C [20,28]. For the CVD SiC coupons, the mass change was almost undetectable from 800 °C to 1200 °C, and at 1300–1350 °C, it showed a mass gain of 2 mg/cm$^2$ [20]. The silica ($SiO_2$) layer on top of the coupon tested for 8 h at 1350 °C was approximately 5 μm thick. The thickness of the silica layer changed depending on the gas velocity, since the oxide volatilization as the compound $Si(OH)_4$ changes with the gas velocity [20]. A large variety of materials were tested during a screening process in a stream of gas containing 50% $H_2O$ in argon at 1 bar pressure and at 1200 °C using thermal gravimetric analysis (TGA). The lowest oxidation rate was for CVD SiC and the second lowest was for APMT [20]. The oxidation rate constants of several materials of interest were plotted as a function of the inverse of the absolute temperature (Arrhenius) to calculate the activation energy [4,28]. The highest oxidation rate constant corresponded to the oxidation of Zircaloy-4 to $ZrO_2$, the lowest oxidation rate constant was for the oxidation of SiC to $SiO_2$, and the second lowest for the oxidation of APMT to $Al_2O_3$. The oxidation activation energy in 100% steam was 84 kJ/mol for Zircaloy-4, 119 kJ/mol for SiC, and 172 kJ/mol for APMT [4].

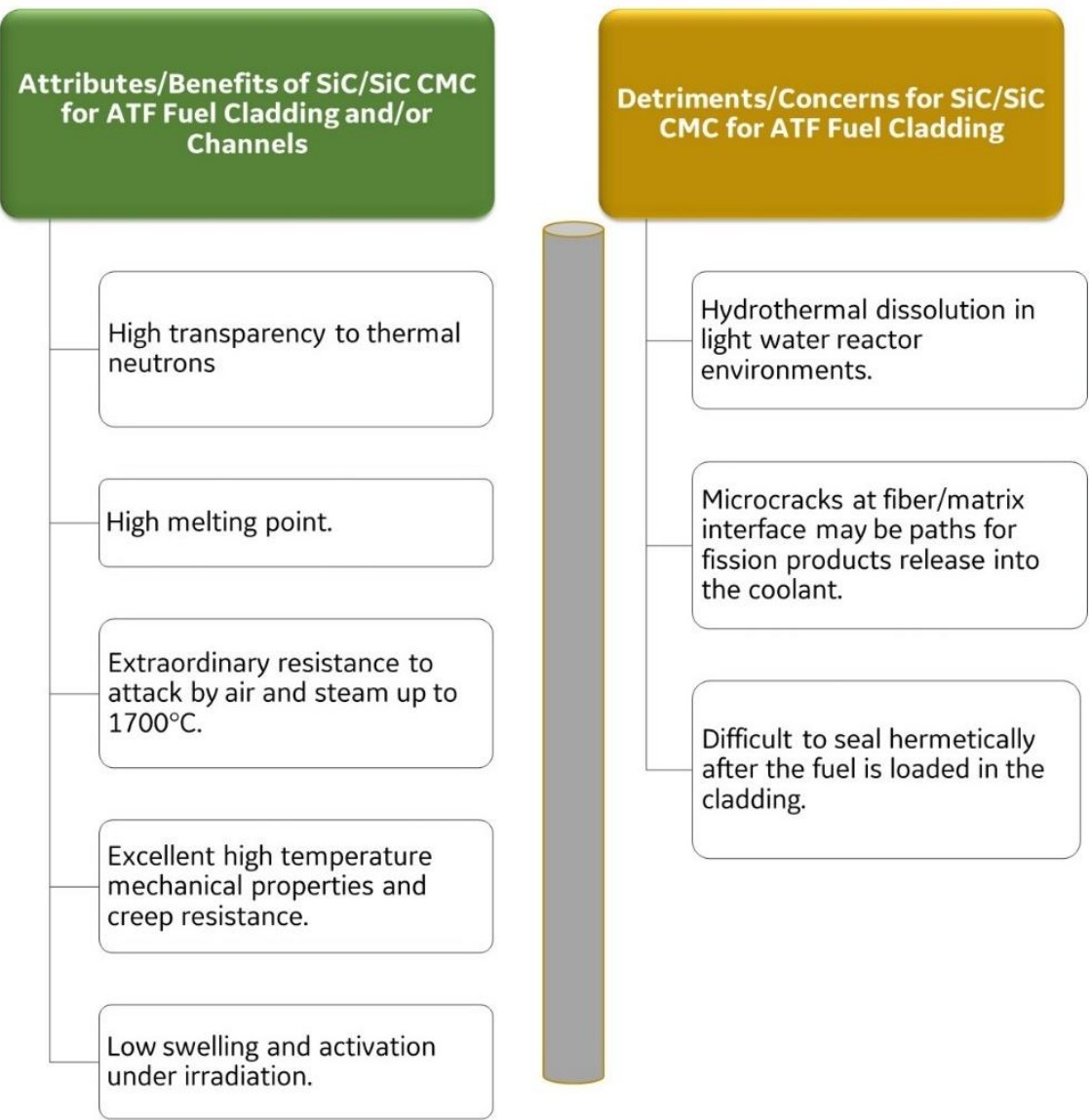

**Figure 7.** Benefits and detriments on the use of SiC composite for an ATF cladding application.

One of the major concerns of the use of SiC composite materials as ATF cladding is their instability in hot water [4,61,63]. The corrosion performance of SiC composites in hot condensed water is a strong function of the fabrication route of the cladding and the exact environmental conditions [60,61]. For LWR applications, the redox potential of the coolant would be an important factor in controlling the recession rate of SiC composites, since the presence of a few ppb of oxygen in the water can be an accelerating factor [61]. In general, researchers report a large scatter of results on the corrosion behavior of SiC-based composites coupons in 300 °C water. Researchers generally use mass change to report degradation (loss or gain). If the material dissolves in the coolant it would represent a mass loss. However, the use of mass loss to account for corrosion is only valid in the case of uniform dissolution (such as for FeCrAl) or uniform oxidation with an adherent oxide (in the case of Zircaloy). For SiC composites, depending on their architecture, corrosion could be rather uniform (such as in the case of matrix CVD SiC), but it could also suffer localized corrosion, in which case evaluating performance by mass change is not recommended [63]. When SiC dissolves in water, the Si forms $SiO_2$ first, and the C may form carbon dioxide ($CO_2$) or methane ($CH_4$) depending on the oxidizing power of the water [4,63]. SiC has exceptional performance in steam systems because silica ($SiO_2$) mainly

stays on the surface, providing further protection as a good barrier for oxygen transport across the oxide. In condensed water systems, the silica does not remain on the surface but immediately dissolves into the coolant as silicic acid (in a similar manner to how carbon dioxide dissolves in water, forming carbonic acid). Since the kinetics of the dissolution of silica in water is faster than the kinetics of the formation of silica from SiC, the recession rate will depend on the initial oxidation rate of SiC; that is, LWR environments with oxidizing species due to irradiation-induced hydrolysis would oxidize SiC faster, and the overall corrosion process would be faster. In systems with hydrogen water chemistry such as in the PWR primary water, the oxidation of SiC will be slower due to the reducing conditions of the coolant, so SiC components may survive better in PWR than in BWR environments.

Immersion corrosion tests of SiC samples were conducted in PWR-type water containing boron and lithium with 50 ccH$_2$/kg at 343 °C for up to 800 days [64]. They reported some scatter on the normalized corrosion results but were still able to determine a clear trend in the behavior. Results reported include an initial faster mass loss (for 50 days) and then the weight of the coupons leveled off.

Microcracks may develop between the impregnated CVD matrix and the fibers due to stresses such as the one produced by a thermal gradient between the surface in contact with the fuel and the surface in contact with the coolant [4,60,61]. These microcracks may be formed at a strain of only 0.1%, and they may be paths for fission products release into the coolant.

One concern that many have raised is the lower-than-Zircaloy-or-FeCrAl thermal conductivity of a composite wall for a SiC cladding. Since the exact final architecture of the cladding is not known, adding layers to the tube wall may decrease the overall thermal conductivity, which would raise the temperature of the fuel [60]. This may jeopardize the intention of having ATF fuels with a high burnup.

Since SiC cannot be welded, a reliable gas-tight sealing mechanism for cladding tubes needs to be developed [61]. This end joint plug must be corrosion- and irradiation-resistant, and its consistent production method in an industrial setting at a reasonable cost still needs to be demonstrated [17,61]. In one architecture of the SiC composite, the use of a tubular metallic liner to provide hermeticity and allow for end caps welding was proposed [17].

One first step on the use of SiC composites for LWR application would be to use them in replacement of the Zr alloys in the BWR channel boxes [4,62]. These boxes are in the reactor core to guide the flow of water along the fuel rods, and they do not require hermeticity since the boxes are open at both ends. The use of SiC in channel boxes will screen the material for interaction with a coolant at a representative flow under neutron irradiation. If the channel boxes made of SiC survives three cycles in a BWR LWR, it would perform well in a PWR application if the sealing plugs and the microcracks hermeticities are resolved. BWR coolant environments are more aggressive to material degradation than PWR primary water environments because of the radiolysis in the BWR water.

Irradiation and Deployment Campaigns for SiC Composites

Rodlets of SiC-composite materials have been prepared for irradiation campaigns [17]. Unfueled tubes are planned to be irradiated at the MIT reactor though an Integrated Research Project (IRP) program. Similarly, fueled rodlets of SiC/SiC are planned to be irradiated at the Advanced Test Reactor (ATR) at the Idaho National Laboratory (INL) [17]. This would be the first step towards SiC-based cladding, allowing for future insertions in commercial reactors as for the other concepts such as coated rodlets and rodlets made with monolithic FeCrAl alloy.

Irradiation exposure tests of SiC tube samples were conducted in the MIT reactor both under gamma irradiation (out-of-core) and under neutron + gamma (in-core) irradiation at 298 °C in water containing 1400 ppm of boron and 4.6 ppm of lithium plus 50 ccH$_2$/kg at a flow rate of 2.5 m/s. The total testing time was 128 effective full-power days [64]. The mass loss was between 1.28% to 2.67%, and it was consistent regardless of in-core and out-of-core positions; therefore, the effect of irradiation on the corrosion behavior is currently inconclusive.

### 2.4. How ATF Can Help Extending the Life of LWRs

Figure 8 shows a schematic representation of the barriers that exist between the toxic components in the fuel cavity (1) and the environment (6). For the radioactive elements in the fuel cavity to be released to the environment, they need to breach the tube or cladding (2), then the water (coolant) (3) in the reactor core, and the last two barriers of the metallic reactor pressure vessel wall (4) and the external cement barrier (5). This type of architecture is what the Nuclear Regulatory Commission calls "defense in depth." In the case of the Fukushima scenario, the lack of water recirculation caused the zirconium alloy cladding tube (2) to react with the surrounded water or steam (v) to produce hydrogen gas and zirconium oxide according to the following:

Zr (s) + 2H$_2$O (l or v) === ZrO$_2$ (s) + 2H$_2$ (g) + Heat of Reaction (highly exothermic).

**Figure 8.** Multiple barriers are built to reduce risk of toxic elements being released in the environment.

In the Fukushima scenario, the cladding of the fuel or barrier (2) reacted with the immediate barrier (3). The plant black-out caused barrier (3) not to respond as designed (i.e., staying liquid and at temperatures below 300 °C). Not only was barrier (3) not effective anymore; it also attacked barrier (2) (letter "R" in Figure 8), causing fuel (1) to be dispersed outside of the cladding containment. The currently proposed ATF concepts such as FeCrAl

are aimed at producing or fabricating more robust barriers (2). In other words, a safer plant operation can be accomplished by minimizing or retiring risks by using ATF claddings that would be less reactive with the coolant. Using ATF materials (such as FeCrAl cladding) may reduce the risk of plant operation. The use of resilient materials will also allow for longer plant operation and extending the periods between refueling, which will also decrease the cost of plant operation. These measures include extending the burnup of the fuel before replacement. Another measure that would decrease the cost of plant operation would be to reduce fuel licensing burdens, especially between different countries.

## 2.5. ATF Materials and the Fuel Cycle

The modified and innovative ATF materials listed in Figure 1 have not yet been in the production of power in an LWR. Figure 9 shows an example of a fuel cycle from cradle to grave. The behavior of Zr-based alloys has been investigated in this entire cycle through the decades. However, the newer ATF materials such as FeCrAl still need to be evaluated. The reliable fabrication of fuel rods (Step 1 in Figure 9) from basic raw materials needs to be demonstrated. The manufacturing of slender seamless tubes of APMT was proven in an industrial setting, using powder metallurgy to HIP ingots, and later applying pilgering up to the final dimensions of 5 m long and 0.3 mm wall thickness. It was also proven that the end caps can be welded to the APMT tubes using pressure resistance welding (no melting). FeCrAl materials are inexpensive and existent tube fabrication plants can manufacture and weld the end caps to the tubes. The fuel rods with innovative materials then need to survive maybe 10 years under normal operation conditions in the reactor while in contact with ~300 °C water under irradiation (Step 2). APMT has been shown to have low recession rates both in oxygenated and hydrogenated water, and the presence of Zn decreases this corrosion even more. If there is a loss of coolant accident, the APMT rods need to resist oxidation and resist the generation of hydrogen gas while maintaining their geometry until the reactor is flooded again with fresh water. After three cycles in the reactor core, the bundles of the APMT rods need to be retrieved in one piece and transported to the cooling pools (Step 3). Residence in the cooling pools could take up to 20 years to allow for the short-lived toxic radionuclides to decay and thus reduce the heat in the rods. The water in the pools is generally at 60 °C, and its chemistry is carefully controlled. For passivating alloys such as APMT in the pool water, the general corrosion should be negligible. Electrochemical tests results showed that APMT was resistant to localized corrosion and that its passivation breakdown potential was higher than that of type 316SS. Step 4 in Figure 9 shows that after the cooling pools, the used fuel would go to dry cask storage, where it would remain for a longer time. In the dry casks, the heat of the bundles is removed by natural air convection, and the temperature of the bundles may never reach 100 °C. Since APMT does not form hydrides that may embrittle the traditional Zr-based alloys, no environmental degradation of APMT clad fuel is anticipated in dry storage. In certain countries, the reprocessing of used fuel is allowed (Step 5 in Figure 9), and there is a method in place for the reprocessing of Zr-based used fuel but no method yet for the newer ATF materials. The dissolution rate of APMT and C26M in mineral acids as a function of the temperature has been tested in out-of-pile laboratory conditions. With or without the option or reprocessing, the long-lived radioactive isotopes are intended, eventually, to be disposed in geologic repositories (Step 6 in Figure 9).

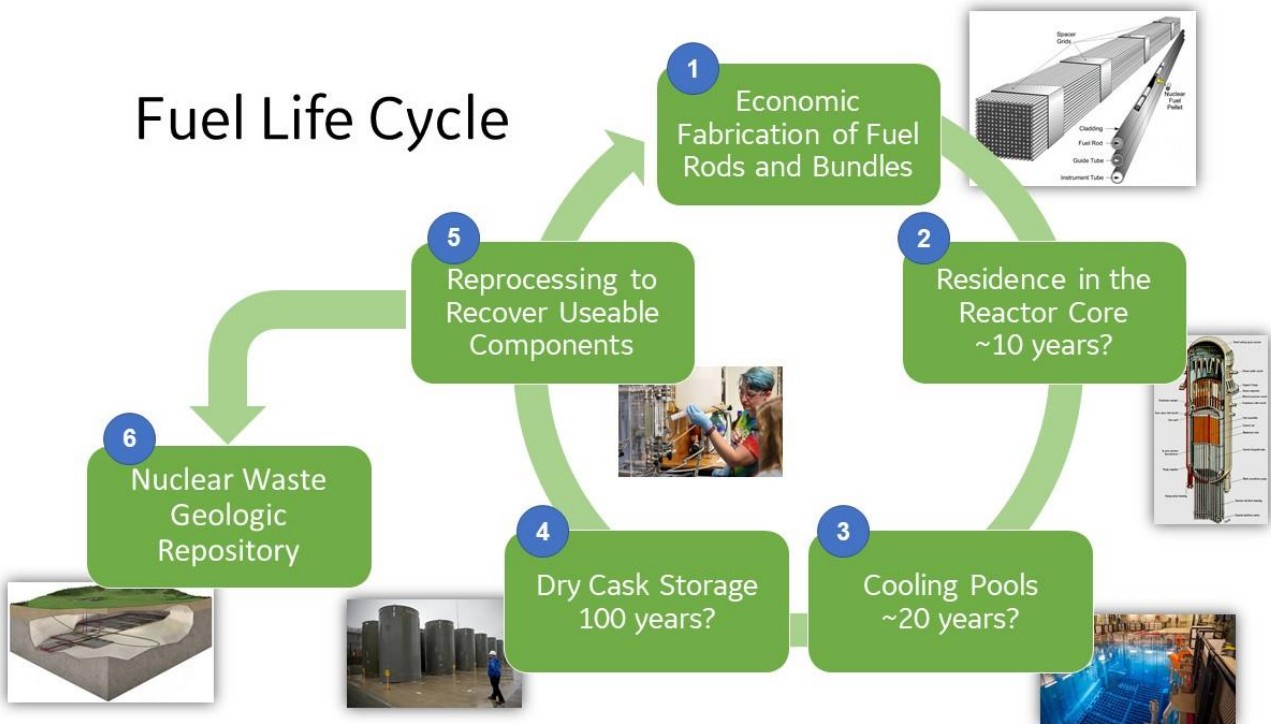

**Figure 9.** Fuel cycle from manufacturing to used fuel disposition.

### 3. Summary and Conclusions

1. Nuclear energy can be used to assist global decarbonization efforts;
2. The decommissioning of existing operational LWRs can be delayed by retrofitting them with more robust accident-tolerant fuels (ATF);
3. There are three main concepts for cladding of the ATF rods: coating existing Zr alloys clads; using monolithic FeCrAl; and using composites based on the ceramic silicon carbide;
4. The most likely ATF materials to be implemented first in commercial reactors would be coated Zr alloys because they will likely need fewer regulatory approval steps;
5. The second most likely ATF concept to be implemented will be FeCrAl alloys if their resistance to neutron irradiation is proven to be acceptable.

**Funding:** This research was funded by the US Department of Energy grant number DE-NE0009047.

**Data Availability Statement:** No new data were created or analyzed in this study. Data sharing is not applicable to this article.

**Conflicts of Interest:** The author declares no conflict of interest.

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
