# Peer review of "Improved and Innovative Accident-Tolerant Nuclear Fuel Materials Considered for Retrofitting Light Water Reactors—A Review"

_cmd, doi:10.3390/cmd4030024_

Round 1

Reviewer 1 Report (New Reviewer)

     The author reviewed the recent development of improved  accident tolerant nuclear fuel materials. He cited many recent studies. This paper will contribute for the understanding of the study of this field. The description of history of nuclear power plants seems to be not necessary. But most of the readers of this Journal are not the specialist of nuclear engineering, and maybe it is no problem. I believe this paper is acceptable for the Journal. But I want to add two points before the publication.

1)The Cr cladding on Zr alloys shows a good performance. But the formation of inter-metallic compound Zr-Cr is one of serious problems. Please add this point in the text.

ï¼’) The reaction of steam and water with SiC is also a serious problem. Some researchers consider that it is impossible to use SiC without some coating. Please write your comment on this point.

Author Response

Reviewer 1

The author reviewed the recent development of improved accident tolerant nuclear fuel materials. He cited many recent studies. This paper will contribute for the understanding of the study of this field. The description of history of nuclear power plants seems to be not necessary. But most of the readers of this Journal are not the specialist of nuclear engineering, and maybe it is no problem. I believe this paper is acceptable for the Journal. But I want to add two points before the publication.

  • The Cr cladding on Zr alloys shows a good performance. But the formation of inter-metallic compound Zr-Cr is one of serious problems. Please add this point in the text.
    Added to the manuscript: Cr may react with Zr to form ZrCr2 intermetallic compounds; however, at the normal operation temperature of the cladding (~350°C) the development of ZrCr2 would not be significant to impair its performance.

  • The reaction of steam and water with SiC is also a serious problem. Some researchers consider that it is impossible to use SiC without some coating. Please write your comment on this point.
    Answer: The liability of SiC about its use in condensed 300°C has been addressed in the manuscript just under Fig. 7.

Reviewer 2 Report (New Reviewer)

The author described the three primary ATF materials in detail and analyzed the development status and prospect of the field. The manuscript is unique and comprehensive. The advantages of each ATF material and the limitations of the current investigations are clearly explained. The author uses accurate and concise words to describe the performance of the Cr coating, FeCrAl and SiC in corrosion resistance, friction and wear resistance. The manuscript is expected to have high impact.

The review can be published after minor revisions:

1.       All images should be cited properly.

2.       Each of the smaller images in Figure 6 should be captioned.

3.       Page 10, line 351, ID should be in abbreviated form. It is better to have the full name.

4.       The summary in this review is a little bit brief with an evaluation of three current ATF materials. It is better to have some discussion on how to further enhance the performance of the current ATF materials.

5.       It is suggested to discuss further the usability of the proposed ATF materials in future nuclear reactors.

Author Response

The author described the three primary ATF materials in detail and analyzed the development status and prospect of the field. The manuscript is unique and comprehensive. The advantages of each ATF material and the limitations of the current investigations are clearly explained. The author uses accurate and concise words to describe the performance of the Cr coating, FeCrAl and SiC in corrosion resistance, friction and wear resistance. The manuscript is expected to have high impact.

Very grateful to Reviewer 2 for the great comments to this manuscript.

The review can be published after minor revisions:

  1. All images should be cited properly.
    Answer: The manuscript was checked that all figures were cited.
  2. Each of the smaller images in Figure 6 should be captioned.
    Answer: The following text was added to caption in Figure 6.
    (a) and (b) A 600 µm deep wear groove formed on the Zirc-2 cladding after two weeks testing in 288°C water. (c) and (d) A shallower 150 µm deep groove formed on the APMT tube after two weeks in 288°C water.
  3. Page 10, line 351, ID should be in abbreviated form. It is better to have the full name.
    Answer: Changed to Internal Diameter.
  4. The summary in this review is a little bit brief with an evaluation of three current ATF materials. It is better to have some discussion on how to further enhance the performance of the current ATF materials.
    Answer: The following text was added to the abstract. “The use of ATF materials may help extending the life of currently operating LWRs, while being a link to materials development for future commercial reactors.”
  5. It is suggested to discuss further the usability of the proposed ATF materials in future nuclear reactors

Answer: The following statement has been added at the end of the section on FeCrAl “Since the FeCrAl ferritic alloys have excellent mechanical properties (Table 3) and oxidation resistance at temperatures in the order of 800°C and higher [Huang et al 2022, Rebak et al 2022] they are also being considered for application in Generation IV reactors and fusion reactors.”

Reviewer 3 Report (New Reviewer)

This paper is summarized the previous studies on the ATF. It is well organized and reflects the current research trend. 

There are no significant modifications, but there are minor modifications that need to be made to improve the quality of the paper, as follows :

 1. The Figures included in the context (Fig. 2, Fig 3, etc.) should be revealed their source 

 2. It is recommended to amend the title of a section

 - 2.2.2: "Resistance to SCC of FeCrAl

 - 2.2.3:  "Passive Corrosion Behavior in normal operation                                      conditions."

 - 2.2.4: "Resistance to debris fretting."

 3. Some section numbers need to be amended.

 - 2.3.2 and 2.3.3 are not the description of the SiC composites, and are general descriptions on the ATF, so their number should be 2.4 and 2.5

Author Response

This paper is summarized the previous studies on the ATF. It is well organized and reflects the current research trend. 

There are no significant modifications, but there are minor modifications that need to be made to improve the quality of the paper, as follows :

  1. The Figures included in the context (Fig. 2, Fig 3, etc.) should be revealed their source 
    Most figures were adapted/modified from previously published ones. A citation was added in the Figure caption for the origin of the figures. All figures shown are by the author.
  2. It is recommended to amend the title of a section

 - 2.2.2: "Resistance to SCC of FeCrAl

 - 2.2.3:  "Passive Corrosion Behavior in normal operation                                      conditions."

 - 2.2.4: "Resistance to debris fretting."
Thank you for your suggestion to shorten the titles of sections 2.2.2, 2.2.3, and 2.2.4. I think the current titles are more descriptive for someone who browses quickly.

  1. Some section numbers need to be amended.

 - 2.3.2 and 2.3.3 are not the description of the SiC composites, and are general descriptions on the ATF, so their number should be 2.4 and 2.5

Excellent comment. Thank you. The numbering was introduced by the editorial office. This numbering was now corrected.

This manuscript is a resubmission of an earlier submission. The following is a list of the peer review reports and author responses from that submission.

Round 1

Reviewer 1 Report

In this paper, a general and qualitative description of the use of novel materials for avoiding the consequences of an accident in light water reactors is included. The analysis is quite general and it could be considered reflections including just general descriptions in the paper instead of a rigorous and deep analysis of the topic as it must be done in a scientific work. Many times in the main text evidences that support the comments are needed and such references are not provided. In addition, references to Wikipedia are not acceptable for a rigorous scientific work. Author sate in the abstract “an overview of the corrosion performance of these revolutionary materials in the entire fuel cycle will be described and ranked”. However, none of these are included in the paper.

To name some other faults of the manuscript:

#1) The sections are not numbered. The citation method is not in accordance to the one included in the journal template.

#2) There is not a common structure of a scientific paper. Introduction (stabilising the state of the art and the aim of the paper), methods and materials, results and discussion and finally conclusions. The first section is rather small and the aim of the paper (an overview of the corrosion performance of such materials) is not included.

#3) Line 14. Introduction section is quite general and it should be reconsidered.

#4) Line 22. Please do not use general and qualitative comments such as “much power”, please include numerical values for the analysis.

#5) Line 25. Please, do not cite Wikipedia

#6) Line 27. Please include evidences of this including references.

#7) Line 27. Please do not include religious terms.

#8) Line 85. Figure and section are only a description of historical facts without discussion.

#9) Line 118. Please include a reference revealing the source from where such data is obtained.

#10) Line 125. Please delete double blank spaces

#11) Line 129. UAE is not defined

#12) Line 131-132- Please provide evidences of this statement.

#13) Line 146. Please avoid large blank spaces after fig. 3 and after fig. 4 (line 158)

#14) Lines 171.173. Please do not include opinions instead of facts.

#15) Line 190-192. Please provide reference for each type

#16) Line 198.Fig. 7 must be numbered correctly; it should be fig. 6

#17) Line 212. Fig. 6 is quite simple. It should be more interesting showing a scheme of a LWR including the components and materials.

#18) Line 224. The acronym OD is defined later in line 252.

#19) Line 254. Please provide evidences of this sentence including references

#20) Line 260. Fig. 9 is just a simple and qualitative analysis

#21) Line 280. Please provide evidences of this including references

#22) Line 291. “(v)” is right?

#23) Line 311. Conclusion section is in fact a summary of the paper with quite general sentences.

#24) Line 324. Reference section should be carefully revised since many format mistakes are easily found. To name some of them:

·        Some references include the author surname and the initial name (ref. 1, the co-authors appears with full name and surname and the first author appears as surname and full name). In other the full name and surname is used and in others the initial name and later the surname is used.  

·        Wikipedia must not be included in references of a scientific work

·        Some references use “and” before the last author and other do not.

·        Do not use et al (ref. 20) considering the 31 co-authors of reference 4.

·        The font of ref. 24 is different.

Reviewer 2 Report

After reading the manuscrit, I find it more close to a news report, not a scientific paper. There are not any new findings or technologies introduced. It might be not appropriate to be published in the journal.